# The Enigmatic Genetic Landscape of Hereditary Hearing Loss: A Multistep Diagnostic Strategy in the Italian Population

**DOI:** 10.3390/biomedicines11030703

**Published:** 2023-02-24

**Authors:** Beatrice Spedicati, Aurora Santin, Giuseppe Giovanni Nardone, Elisa Rubinato, Stefania Lenarduzzi, Claudio Graziano, Livia Garavelli, Sara Miccoli, Stefania Bigoni, Anna Morgan, Giorgia Girotto

**Affiliations:** 1Department of Medicine, Surgery and Health Sciences, University of Trieste, 34149 Trieste, Italy; 2Institute for Maternal and Child Health—I.R.C.C.S. “Burlo Garofolo”, 34137 Trieste, Italy; 3U.O. Genetica Medica, AUSL della Romagna, 47521 Cesena, Italy; 4Medical Genetics Unit, Azienda USL-IRCCS di Reggio Emilia, 42122 Reggio Emilia, Italy; 5Unit of Medical Genetics, Azienda Ospedaliero Universitaria di Bologna, IRCCS, 40126 Bologna, Italy; 6Medical Genetics Unit, Department of Mother and Child, Ferrara Sant’Anna University Hospital, 44124 Ferrara, Italy

**Keywords:** hereditary hearing loss, MLPA, long-range PCR, whole exome sequencing, non-syndromic mimics, dual molecular diagnosis

## Abstract

Hearing loss is the most frequent sensorineural disorder, affecting approximately 1:1000 newborns. Hereditary forms (HHL) represent 50–60% of cases, highlighting the relevance of genetic testing in deaf patients. HHL is classified as non-syndromic (NSHL—70% of cases) or syndromic (SHL—30% of cases). In this study, a multistep and integrative approach aimed at identifying the molecular cause of HHL in 102 patients, whose *GJB2* analysis already showed a negative result, is described. In NSHL patients, multiplex ligation probe amplification and long-range PCR analyses of the *STRC* gene solved 13 cases, while whole exome sequencing (WES) identified the genetic diagnosis in 26 additional ones, with a total detection rate of 47.6%. Concerning SHL, WES detected the molecular cause in 55% of cases. Peculiar findings are represented by the identification of four subjects displaying a dual molecular diagnosis and eight affected by non-syndromic mimics, five of them presenting Usher syndrome type 2. Overall, this study provides a detailed characterisation of the genetic causes of HHL in the Italian population. Furthermore, we highlighted the frequency of Usher syndrome type 2 carriers in the Italian population to pave the way for a more effective implementation of diagnostic and follow-up strategies for this disease.

## 1. Introduction

The primary aim of medical genetics is the identification of the molecular cause of rare diseases, which, despite being singularly rare, may affect up to 8% of newborns [1]. However, recognising the specific genetic condition that affects a patient is anything but easy, since the number of rare disorders is exceptionally high and many of them are characterised by a significant clinical and genetic heterogeneity, as in hereditary hearing loss (HHL).

Congenital hearing loss is the most frequent sensorineural disorder, with a prevalence of approximately 1:1000 live births [2]. In developed countries, 50% to 60% of all cases are due to genetic causes (https://www.cdc.gov/ncbddd/hearingloss/genetics.html, accessed on 25 January 2023) and are referred to as HHL. HHL can be classified as non-syndromic (NSHL), when deafness is the only present sign, or syndromic (SHL), when hearing loss (HL) is accompanied by other clinical features [3]. NSHL accounts for approximately 70% of HHL cases and, so far, more than 120 genes have been associated with this condition. Among them, 51 are causative of autosomal dominant (AD) forms, 77 of autosomal recessive (AR) forms, and five of X-linked (XL) forms; notably, some genes are responsible for both AD and AR NSHL, additionally highlighting the complexity of inheritance patterns associated with this condition (Hereditary Hearing Loss Homepage; https://hereditaryhearingloss.org/, last accessed on 25 January 2023). According to the literature, the major players involved in NSHL are the *GJB2* gene, which accounts for approximately 50% of all AR cases, and *STRC*, which is increasingly recognized as the second-most significant contributor to AR NSHL [4]. On the other hand, SHL represents 30% of cases of HHL and about 400 syndromes that include this phenotype have been reported in the literature [5]. Some of them account for a substantial fraction of cases, such as Pendred, Usher, Alport, and Waardenburg syndromes [6], whereas others are very rare, such as Perrault and Barakat syndromes [7,8].

In this entangled genetic landscape, defining the molecular cause of HHL poses a real challenge to diagnosticians, since there are further peculiar circumstances to take into consideration. For instance, it has recently been demonstrated that the presence of a dual molecular diagnosis in HHL patients is a particularly frequent event that needs to be suspected whenever signs and symptoms do not fit a known syndromic pattern or whenever there is a high intrafamilial clinical variability [9]. Another occurrence that complicates HHL diagnosis is represented by non-syndromic mimics: they are defined as syndromic forms of HL that masquerade as NSHL, since additional clinical features either are extremely subtle or develop later in life [3,10].

In this light, the optimisation of high-throughput sequencing technologies, such as whole exome sequencing (WES), together with complementary genetic diagnostic approaches, such as multiplex ligation probe amplification(MLPA) and long-range PCR (LR-PCR) have enormously improved molecular and clinical geneticists’ ability to define the precise molecular diagnosis of HHL patients. Indeed, the possibility to simultaneously analyse thousands of genes with integrated approaches is the cornerstone to face HHL’s extensive genetic and clinical heterogeneity.

In this context, we hereby report the results of our study aimed at the identification of the genetic cause of HHL in 102 patients recruited in the last 18 months and analysed through a multistep and integrative approach. The final goal of this study is to provide a detailed characterisation of the genetic causes of HHL other than *GJB2* mutations in the Italian population, with particular attention to the identification of peculiar scenarios such as the presence of dual molecular diagnoses or non-syndromic mimics.

## 2. Materials and Methods

### 2.1. Ethical Statement

All the analyses were performed following the relevant guidelines and regulations. Written informed consent was obtained from all participants or, in the case of underage patients, their legal guardians. The study was conducted in accordance with the tenets of the Helsinki Declaration and was approved by the Ethics Committee of the I.R.C.C.S. “Burlo Garofolo” of Trieste.

### 2.2. Study Design

In the last 18 months, 102 patients affected by NSHL and SHL and negative on *GJB2*, *GJB6*, and *MT-RNR1* first-tier screenings were referred to the Medical Genetics Unit of the I.R.C.C.S. “Burlo Garofolo” (Trieste, Italy), the Medical Genetics Unit of the “AUSL della Romagna” (Cesena, Italy), the Medical Genetics Unit of the “Azienda USL—I.R.C.C.S. di Reggio Emilia” (Reggio Emilia, Italy), the Medical Genetics Unit of the I.R.C.C.S. Azienda Ospedaliero-Universitaria “Policlinico di Sant’Orsola” (Bologna, Italy), or the Medical Genetics Unit of the “Sant’Anna” University Hospital (Ferrara, Italy).

The enrolled subjects were analysed through a multistep approach that comprises the following steps: (1) a detailed clinical evaluation to distinguish NSHL patients from SHL ones; (2) MLPA analysis of *STRC-CATSPER2* and *OTOA* in NSHL patients; (3) LR-PCR and *STRC* sequencing in patients carrying a heterozygous deletion of *STRC* and an audiometric pattern suggestive of Deafness, autosomal recessive 16; (4) WES analysis in patients negative to steps 2 and 3 and in SHL subjects.

Regarding the clinical evaluation, all participants were deeply characterised through a detailed anamnesis, a dysmorphological examination, an audiological evaluation, and further investigations. In particular, a complete familial anamnesis was collected in order to identify possible other affected family members and a personal medical history was obtained to highlight potential confounding factors (e.g., infections, trauma, or other non-genetic causes of HL). A careful physical examination was carried out to identify possible dysmorphic features, with particular attention to the overall appearance and to facial, ectodermal, skeletal, and genital features. All patients underwent pure tone audiometry testing (PTA) or auditory brainstem response (ABR), according to the proband’s age, to define the degree of HL [11]. Ancillary clinical tests included brain magnetic resonance imaging (MRI) and computed tomography (CT), thyroid function assessment, abdominal ultrasound, and neurological, ophthalmological, and cardiological evaluations.

### 2.3. DNA Extraction and Quality Control

Genomic DNA was extracted from peripheral whole-blood samples of patients and both their parents using the QIAsymphony^®^ SP instrument with QIAsymphony^®^ Midi Kit (Qiagen, Venlo, The Netherlands), following the manufacturer’s instructions. The DNA quality was verified through 1% agarose gel electrophoresis and the DNA concentration was measured using the Nanodrop ND 1000 spectrophotometer (NanoDrop Technologies Inc., Wilmington, DE, USA).

### 2.4. MLPA

MLPA analysis for the identification of deletions and duplications involving the *STRC-CATSPER2* and *OTOA* genes was performed using SALSA^®^ MLPA^®^ probe mixes P461-B1 DIS (MRC-Holland, Amsterdam, the Netherlands. The kit contains different probes spanning the aforementioned genes. In particular, seven probes for the *STRC* gene (covering exons 19, 20, 23, 24, 25, and 28), five probes for the *CATSPER2* gene (covering exons 1, 2, 4, and 7), and ten probes for the *OTOA* gene (covering exons 2, 5, 7, 8, 11, 12, 16, 17, 18, and 20) are present. Multiple flanking probes are also included in the 15q15.3 and 16q12.2 regions, indicating the extent of possible copy number variations (CNVs). Furthermore, four *STRC* pseudogene (p-*STRC*) probes are present to identify possible gene conversions. Samples were performed according to the manufacturer’s instructions. Briefly, 50–100 ng of DNA was denatured and hybridised overnight with the SALSA^®^ probe mix; afterwards, samples were treated with DNA ligase and polymerase chain reaction (PCR) amplification was carried out with specific fluorescent-labelled PCR primers. Amplified product fragment analysis was performed on an ABI 3500dx Genetic Analyzer (Life-Technologies, Carlsbad, CA, USA). Coffalyser.Net software v.220513.1739 (MRC Holland - Amsterdam, the Netherlands) was employed for data analysis in combination with the lot-specific MLPA Coffalyser sheet. The following cut-offs for dosage quotient (DQ) of the probes were applied to interpret the MLPA results: 0.80 < DQ < 1.20 (no deletion/duplication), DQ = 0 (homozygous deletion), 0.40 < DQ < 0.65 (heterozygous deletion), 1.30 > DQ > 1.65 (heterozygous duplication), and 1.75 < DQ < 2.15 (homozygous duplication/heterozygous triplication).

### 2.5. LR-PCR and STRC Sequencing

LR-PCR was performed on all patients carrying a heterozygous *STRC* deletion and presenting an audiometric pattern suggestive of Deafness, autosomal recessive 16 (MIM: # 603720), due to biallelic variants in the *STRC* gene. Considering the presence of a highly homologous p-*STRC*, in order to avoid p-*STRC* sequences, two LR-PCR products were generated for subsequent nested PCR using the UltraRun LongRange PCR Kit (Qiagen, Venlo, The Netherlands), as described by Vona et al. [12]. In particular, two LR-PCR fragments, the first from exon 1 to exon 19 and the second from exon 12 to exon 29, were obtained. They both overlap intron 18, which is a non-identity region, where only *STRC* presents three additional nucleotides. After amplification, a 1% agarose gel was employed to verify the fragments’ length and PCR products were diluted 1:1000 to decrease pseudogene carryover. Intron 18 was therefore amplified and sequenced to confirm pseudogene exclusion. After ensuring the exclusive amplification of *STRC*, nested PCRs and Sanger sequencing of all exons were performed.

### 2.6. WES and Data Analysis

WES was performed on an Illumina NextSeq 550 instrument (Illumina Inc., San Diego, CA, USA) using the Twist Human Core Exome and Human RefSeq Panel kit (Twist Bioscience, South San Francisco, CA, USA) according to the manufacturer’s protocol. Briefly, genomic DNA was enzymatically fragmented, ligated to a universal adapter, and amplified using the Unique Dual Index primers (Twist Bioscience, South San Francisco, CA, USA). Samples were therefore hybridised with the Twist Human Core Exome and the Human RefSeq Panel kit, which cover 99% of protein-coding genes. Hybridised fragments were captured, amplified, and sequenced.

The process allows the production of FASTQ files that were analysed through a custom pipeline (Germline-Pipeline) developed by enGenome s.r.l. (https://www.engenome.com/, Pavia, Italy, accessed on 25 January 2023). This pipeline comprises several steps, such as FASTQ trimming, FASTQ quality check, FASTQ mapping, marking of duplicates, base quality score recalibration, and variant calling, thus permitting the identification of germline variants, including single nucleotide variants (SNVs), short insertions/deletions (INDELs), and exon-level CNVs starting from sequence reads. The secondary analysis, therefore, leads to the generation of final VCF files that contain SNVs, INDELs, and CNVs.

VCF files were analysed through the enGenome Expert Variant Interpreter (eVai) software (https://evai.engenome.com/, Pavia, Italy, accessed on 25 January 2023), which allows variant annotation, interpretation, and prioritisation. In particular, eVai completes the prioritisation process by exploiting both artificial intelligence and the American College of Medical Genetics and Genomics/Association for Molecular Pathology (ACMG/AMP) guidelines to analyse and classify genomic variants [13].

The variant frequency was verified both in NCBI dbSNP build 155 (https://www.ncbi.nlm.nih.gov/snp/, accessed on 25 January 2023) and gnomAD (https://gnomad.broadinstitute.org/, accessed on 25 January 2023) in order to exclude variants previously reported as polymorphisms. The pathogenicity of already-reported variants was assessed through the Human Gene Mutation Database^®^ (HGMD^®^) (https://my.qiagendigitalinsights.com/bbp/view/hgmd/pro/start.php, accessed on 25 January 2023), Deafness Variation Database (http://deafnessvariationdatabase.org/, accessed on 25 January 2023), and ClinVar (https://www.ncbi.nlm.nih.gov/clinvar/, accessed on 25 January 2023). All databases were last accessed on 25 January 2023. The effect of all identified variants was evaluated through in silico prediction tools, such as PolyPhen-2 [14], Sorting Intolerant From Tolerant (SIFT) [15], Pseudo Amino Acid Protein Intolerance Variant Predictor (PaPI score) [16], Deep Neural Network Variant Predictor (DANN score) [17], and dbscSNV score [18]. SNVs leading to synonymous amino acid substitutions not predicted as damaging, not affecting splicing, or highly conserved residues were excluded; variants with a quality score (QUAL) < 20 or called in off-target regions were excluded as well.

Variants were discussed within a multidisciplinary team to assess whether they could be possibly matched to each patient’s phenotype; all variants of interest were confirmed with Sanger sequencing. Familial segregation of the identified variants was also performed using Sanger sequencing.

### 2.7. Prevalence of USH2A and ADGRV1 Pathogenic Variant Carriers in Italian Cohorts

One thousand two hundred eleven individuals from three different Italian cohorts were included in this analysis: (1) the Friuli-Venezia Giulia (FVG) cohort, which includes a collection of samples from six villages (Clauzetto, Erto, Illegio, Resia, San Martino del Carso, and Sauris) located in northeastern Italy [19]; (2) the Val Borbera (VBI) cohort, which consists of samples collected in a geographically isolated valley in the northwest of Italy [19]; (3) the Carlantino (CAR) cohort, which collects samples from a small village located in the Puglia region in southern Italy [19]. Whole genome sequencing (WGS) and detailed phenotypic data of these subjects were already available as an in-house database as a result of previous studies [20]. Only healthy individuals with available WGS data were considered; specifically, 663 individuals were selected from the FVG cohort, 424 from the VBI cohort, and 124 from the CAR cohort. WGS data were generated and analyzed in each cohort separately, as previously described by Cocca et al. [19].

*USH2A* and *ADGRV1* gene variants were extracted considering a minor allele frequency of 5%. Functional annotation was performed using the Variant Effect Predictor tool [21] and frameshift, splice acceptor, splice donor, start lost, stop gained, and stop lost variants were extracted from the generated data using bcftools’ plug-in “Split-VEP” [22]. The pathogenicity of the extracted variants was assessed through HGMD^®^.

The frequency of the extracted variants was checked in Non-Finnish Europeans in the GnomAD database (https://gnomad.broadinstitute.org/, accessed on 25 January 2023). A sample test of proportions was performed to compare the pathogenic variant carrier frequency of the *USH2A* and *ADGRV1* genes in our cohorts with the prevalence reported in the literature. The statistical significance was set to *p*-value < 0.05. This analysis was performed with R version 4.1.2 (R Foundation for Statistical Computing, Vienna, Austria).

## 3. Results

In the last 18 months, 102 patients affected by HHL and negative in *GJB2*, *GJB6* and *MT-RNR1* analyses were recruited and analysed through a multistep and integrative approach (Figure 1) that comprises: (1) a detailed clinical characterisation to distinguish NSHL patients from SHL ones; (2) the analysis of *STRC-CATSPER2* and *OTOA* deletions in patients affected by NSHL; (3) *STRC* sequencing in patients carrying a heterozygous deletion of *STRC* and an audiometric pattern suggestive of Deafness, autosomal recessive 16; (4) WES analysis in patients negative to steps 2 and 3 and in SHL subjects. The clinical evaluation allowed the identification of 82 patients affected by apparent NSHL and 20 presenting SHL.

### 3.1. STRC Analysis

All NSHL patients first underwent MLPA analysis of *STRC-CATSPER2* and *OTOA* genes in order to identify possible deletions. Six patients (7.3%) carried a homozygous deletion of the entire *STRC* gene and one patient (Patient 3) presented an entire *STRC* gene deletion in compound heterozygosity with a deletion involving only exon 19 of the same gene; none of the analysed subjects presented a biallelic deletion involving *OTOA* (Table 1).

Additionally, six patients (Patients 8, 9, 10, 11, 12, and 13) were carriers of a heterozygous deletion of *STRC*. Parental segregation of the deletion showed that it had been inherited from the father in four cases and from the mother in the remaining two (Table 2). Five of them (Patients 8, 9, 10, 11, and 12) had previously undergone PTA and their bone conduction audiograms are reported in Figure 2: they all presented an audiometric pattern compatible with Deafness, autosomal recessive 16 (MIM: # 603720), which is characterised by mild-to-moderate bilateral hearing impairment [4]. An ABR test had been performed on one patient (Patient 13) due to her age and revealed a threshold compatible with moderate HL. In these patients, in consideration of their clinical features and the suggestive presence of a heterozygous *STRC* deletion, LR-PCR analysis of the *STRC* gene (NM_153700.2) was performed with the final aim of identifying a possible hemizygous in trans variant that could explain their clinical phenotype. Indeed, in all of them, a hemizygous variant was identified and parental segregation showed that it had been maternally inherited in four of them and paternally in the last two cases, thus confirming their presence in trans with the previously identified gene deletion and allowing the achievement of a molecular diagnosis in an additional 7.3% of cases (Table 2).

### 3.2. WES Analysis in NSHL Patients

Sixty-nine patients affected by apparent NSHL and negative in steps 2 and 3 of the diagnostic workflow underwent in trio WES analysis. A molecular diagnosis was confirmed in 26 cases (37.9%) (Table 3).

Among NSHL patients that received a molecular diagnosis, 10/26 were affected by an isolated AR form of HL, underlining how recessive HL is the most common hereditary deafness, in line with previous studies reported in the literature [23]. These cases are represented by Patients 16, 19, 20, 21, 24, 27, 29, 32, 38, and 39. Alongside the frequent forms of AR NSHL, such as *MYO15A-* and *TMPRSS3*-related [24], variants in rarely mutated genes were also identified, as the cases of Patient 21, carrying a homozygous splicing variant in the *LRTOMT* gene, and Patient 27, presenting two compound heterozygous variants in the *CLDN14* gene. Additionally, 2/26 patients were carriers of biallelic variants in *OTOF*, which is associated with auditory neuropathy, autosomal recessive 1 (MIM: # 601071). Conversely, only 4/26 patients were affected by AD NSHL, due to heterozygous variants in the *CECAM16*, *MYH14*, *KCNQ4*, and *PLS1* genes, respectively.

Furthermore, WES analysis allowed the identification of two interesting events: the presence of eight cases of non-syndromic mimics and the occurrence of two patients affected by a dual molecular diagnosis.

Among subjects affected by non-syndromic mimics, Patients 15, 17, and 34 may be included. Patient 15 presents a *de novo* heterozygous variant in the *GATA3* gene, whose pathogenic variants are associated with hypoparathyroidism, sensorineural deafness, and renal dysplasia syndrome (MIM: # 146255). Currently, the five-year-old girl only shows HL; no signs of renal dysplasia are visible upon abdominal ultrasound and calcium and parathormone levels are within the reference range. Patient 17 carries two compound heterozygous variants in the *HARS2* gene, which is associated with Perrault syndrome 2 (MIM: # 614926). The patient, a two-year-old girl, currently presents only HL. Additionally, Patient 34, a 17-month-old boy whose HL was the only feature detected during the initial clinical evaluation, presents a known [25] heterozygous *de novo* pathogenetic variant in the *MITF* gene, causative of Waardenburg syndrome type 2a (MIM: # 193510). Furthermore, this cohort was strikingly enriched in patients affected by Usher syndrome type 2 that at present only show HL and have not developed retinitis pigmentosa yet. Indeed, they represent 5/26 (19.2%) of the analysed subjects, namely Patients 30 and 33, who are carriers of biallelic variants in the *ADGRV1* gene, and Patients 35, 36, and 37, carriers of biallelic variants in the *USH2A* gene.

Patients 14 and 26 were affected by a dual molecular diagnosis: both of them carry a heterozygous variant in a gene associated with NSHL (respectively, *ATP2B2*—Deafness, autosomal dominant 82, MIM: # 619804, and *COL11A1*—Deafness, autosomal dominant 37, MIM: # 618533) and a heterozygous variant in a gene associated with Waardenburg syndrome (respectively, *EDN3*—Waardenburg syndrome type 4, MIM: # 613265, *PAX3*—Waardenburg syndrome type 1, MIM: # 193500).

Finally, WES analysis of Patient 39, a 6-year-old boy affected by bilaterally symmetric moderate sensorineural HL, highlighted the presence of two in trans variants in the *STRC* gene. In consideration of this result, LR-PCR analysis was performed to overcome the p-*STRC* issue and, indeed, it confirmed the molecular diagnosis, further corroborated by the clinical phenotype of the proband.

### 3.3. WES Analysis in SHL Patients

Twenty patients affected by SHL underwent WES analysis and a molecular diagnosis was identified in 11 of them (55%) (Table 4).

Five out of 11 patients that were molecularly diagnosed were affected by Usher syndrome type 1 (Patients 44, 48, and 49) or type 2 (Patients 41 and 45); overall, the prevalence of Usher syndrome among the 20 analysed syndromic patients was 25%, confirming the high frequency of this condition in HHL cohorts. Two patients were diagnosed with relatively frequent forms of SHL and both were already suspected upon the initial clinical evaluation: Patient 40 was affected by Charge syndrome (MIM: # 214800) and Patient 43 by branchiootorenal syndrome 1 (MIM: # 602588).

Additionally, two patients were carriers of heterozygous variants in two recently described genes. In particular, Patient 42 presented a heterozygous stop-gain variant in the *SF3B2* gene, whose pathogenic variants were reported in 2021 to be causative of craniofacial microsomia (MIM: # 164210) [26]. Indeed, Patient 42 showed facial asymmetry, mandibular hypoplasia, asymmetric ear abnormalities resembling a question mark ear, bilateral preauricular tags, and a pit along the branchial arch, together with bilateral conductive hearing loss, thus matching the phenotype of the other described patients. Furthermore, Patient 47 carried a heterozygous splicing variant in the *PPP1R12A* gene, which was described in 2020 as mutated in subjects affected by genitourinary and/or/brain malformation syndrome (MIM: # 618820) [27]. This condition is characterised by abnormal internal and external genitalia, structural renal abnormalities, brain malformations, and eye and skeletal anomalies. Patient 47 showed right megaureter and ureterocele, unilateral myopia of the right eye, and symmetric profound sensorineural HL; the latter has not been reported in any of the twelve patients described in the literature.

Finally, among SHL patients, two subjects affected by a dual molecular diagnosis were also identified. Patient 46, affected by congenital bilateral profound sensorineural HL and retinitis pigmentosa, was a carrier both of the most frequent variants associated with NSHL (c.35delG, p.(Gly12Valfs * 2) in the *GJB2* gene [28]) and of a novel heterozygous frameshift variant in the *RPGR* gene, which is causative of X-linked retinitis pigmentosa 3 (MIM: #300029). WES analysis of Patient 50, presenting severe bilateral sensorineural HL and cutaneous signs of pseudoxanthoma elasticum, highlighted the presence of a known heterozygous variant in the *WFS1* gene [29], associated with Deafness, autosomal dominant 6/14/38 (MIM: # 600965) and an already-reported homozygous variant in the *ABCC6* gene [30], causative of pseudoxanthoma elasticum (MIM: # 264800).

### 3.4. Prevalence of USH2A and ADGRV1 Pathogenic Variant Carriers in Italian Cohorts

A noteworthy feature of this study is represented by a peculiar enrichment in patients affected by Usher syndrome type 2, carrying biallelic variants in the *USH2A* (4/7) and *ADGRV1* (3/7) genes.

The estimated frequency of pathogenic variant carriers in the *USH2A* gene is approximately 1:70 in the general population [31] and no data are available regarding *ADGRV1* pathogenic variant carriers. Furthermore, the prevalence of *USH2A* and *ADGRV1* pathogenic variant carriers in the Italian population is unknown. In this light, the carrier frequency of pathogenic variants within the *USH2A* and *ADGRV1* genes in the Italian population was evaluated, taking advantage of WGS data of 1211 individuals from three different Italian cohorts, namely FVG, VBI, and CAR. Pathogenic variants within *USH2A* and *ADGRV1* genes were extracted and checked through HGMD^®^. The complete list of the extracted variants for each cohort is reported in Appendix A.

Regarding *USH2A*, heterozygous pathogenic variants were detected in 22/663 (3.3%) individuals of the FVG cohort, in 3/424 (0.70%) subjects of the VBI cohort, and in 8/124 (6.5%) individuals belonging to the CAR cohort. In these Italian cohorts, the overall carrier frequency of pathogenic variants in *USH2A* was 33/1211 (2.7%). Concerning the *ADGRV1* gene, heterozygous pathogenic variants were identified in 23/663 (3.5%) individuals of the FVG cohort, in 5/424 (1.2%) subjects of the VBI cohort, and in 3/124 (2.4%) individuals belonging to the CAR cohort. The overall prevalence of *ADGRV1* pathogenic variant carriers in these Italian cohorts was 31/1211 (2.6%). To compare the frequency of *USH2A* pathogenic variant carriers in our cohorts with the available literature data, a sample test of proportions was performed. The statistical analysis result revealed that the frequency of pathogenic variant carriers in the *USH2A* gene in these Italian cohorts was statistically significantly higher (*p*-value = 0.0001436) with respect to the available literature data [31].

## 4. Discussion

The multistep and integrated diagnostic workflow described in this study was an effective tool to provide HHL patients with a molecular diagnosis. Indeed, regarding NSHL patients, *STRC* MLPA and LR-PCR analyses allowed us to solve 13 cases, while WES explained 26 additional ones: overall, the detection rate of this approach was 47.6%. Concerning SHL subjects, WES analysis was fundamental to highlight the molecular cause in 55% of patients. These results are in line with previous studies, from both Italian cohorts and international ones, that verified how the overall diagnostic yield in HHL patients is around 50% [32,33,34,35].

The *STRC* gene turned out to be a significant contributor to AR NSHL also in this cohort, as previous works already reported [4]; in particular, the incidence of *STRC* deletions has been estimated at between 1% and 5% in HL patients [36]. Conversely, *STRC* SNVs have been reported only in a few cohorts [12] due to the complexity of interpretation of *STRC* sequencing data secondary to the presence of p-*STRC*. Despite the fact that a detailed genotype–phenotype correlation between *STRC* loss-of-function and mild-to-moderate HL has been reported in studies involving a limited number of participants, this correspondence seems extremely consistent in all of them [4,37,38,39]. In this light, diagnostic tools able to highlight point mutations in this gene overcoming the pseudogene issue should be implemented in all laboratories specialised in the diagnostic work-up of HHL. Indeed, in our cohort, LR-PCR analysis in six patients with specific audiometric features (i.e., mild-to-moderate HL) and the suggestive presence of a heterozygous *STRC* deletion led to the identification of an in trans hemizygous variant in all of them. Additionally, this diagnostic technique confirmed the presence of the two SNVs in the *STRC* gene detected by WES analysis in Patient 39. This finding, in particular, underlines how the possibility to perform *STRC* sequencing in all patients with mild-to-moderate HL, even without a heterozygous *STRC* deletion, should be considered to increase the diagnostic yield in this specific subgroup of HHL patients.

Concerning WES analysis in both NSHL and SHL patients, the results of the present study confirm the role of few genes as major players in both categories, such as *MYO15A* and *TMPRSS3* in NSHL patients [24] and *CDH23* in SHL ones [40].

Within NSHL subjects, patients affected by non-syndromic mimics represented 8/26 cases (30.8%). Non-syndromic mimics may be divided into two categories: (1) patients that present subtle signs and symptoms attributable to the identified syndrome that were missed during the initial clinical evaluation; (2) patients that currently present only one sign/symptom and may develop other characteristics later in life. Within the first group, Patient 34, a carrier of a known heterozygous variant in the *MITF* gene, might be included: during the initial clinical evaluation, only HL was detected, but he will undergo further assessment upon readmission to identify possible subtle signs of Waardenburg syndrome, such as mild pigmentary abnormalities of the skin, eyes, and hair. Conversely, in the second group, Patient 15, a carrier of a novel heterozygous variant in the *GATA3* gene associated with hypoparathyroidism, sensorineural deafness, and renal dysplasia syndrome, will need to be constantly monitored over time to unveil possible early signs of hypoparathyroidism. Patient 17, presenting two compound heterozygous novel variants in the *HARS2* gene causative of Perrault syndrome, also belongs to the second group and clinical features such as primary amenorrhea and infertility would only be evaluated when she reaches the pubertal stage.

Additionally, a peculiar finding of this study is represented by the overall identification of seven patients (7/102—6.9%) affected by Usher syndrome type 2 due to biallelic variants in the *USH2A* (4/7) and *ADGRV1* (3/7) genes. These patients have been identified both among subjects presenting with apparent NSHL (5/7) and within SHL patients (2/7). In the NSHL group, they qualify as non-syndromic mimics [10], as the identified subjects currently present only HL and have not yet developed retinitis pigmentosa, which is an age-dependent feature. In these cases, the identification of the correct molecular diagnosis before the onset of vision loss is fundamental to prompt a specific clinical follow-up. For instance, periodic ophthalmological evaluations should be planned and, whenever patients develop severe progressive HL with poor speech discrimination and communication difficulties even with hearing aids, making these children excellent cochlear implant candidates, surgery should be performed before the onset of ocular problems to maximise communication skills [41]. In the SHL group, Patient 41 came to the geneticist’s attention at 48 years of age and had already developed vision loss together with HL: in this case, the clinical diagnosis was straightforward and thus confirmed by the molecular one. Conversely, Patient 45, a 5-year-old boy, presented HL and mild facial dysmorphisms, such as frontal bossing and dermal translucency of the head: in this subject, the identification of the molecular diagnosis was able to explain his HL and prompt the appropriate ophthalmological follow-up, but was not able to fully explain his facial features.

The prevalence of Usher syndrome type 2 patients in our cohort is slightly higher than expected [31] and, to date, no published study has so far analysed the prevalence of *USH2A* and *ADGRV1* pathogenic variant carriers in the Italian population. Knowing the exact prevalence of Usher syndrome type 2 carriers within a specific ethnic group is fundamental for physicians to plan tailored healthcare intervention in everyday clinical practice. In this light, the frequency of *USH2A* and *ADGRV1* pathogenic variant carriers in the Italian population was evaluated, taking advantage of already available in-house WGS data of 1211 healthy individuals belonging to three different Italian cohorts. Bioinformatic analyses revealed that the frequency of pathogenic variant carriers in the Italian population is around 2:70, for both the *USH2A* and *ADGRV1* genes. Notably, the frequency of *USH2A* pathogenic variant carriers was statistically significantly higher (*p*-value < 0.05) compared with previous studies [31]. It has to be considered that the available literature data about *USH2A* pathogenic variant carrier frequency in the worldwide population is limited and not updated. In this light, further epidemiological data will be needed to accurately evaluate the prevalence of *USH2A* pathogenic variant carriers across different ethnic populations. Additionally, for the first time, the prevalence of *ADGRV1* variant carriers in the Italian population was provided. These data could be considered as a starting point for a better and tailored clinical management: indeed, providing a more accurate picture of Usher syndrome type 2 carrier frequency could contribute, in a long-term perspective, to a more effective implementation of diagnostic and follow-up strategies in the clinical practice.

Finally, in both NSHL and SHL patients, two cases of dual molecular diagnoses have been identified: overall, four out of 89 patients (4.5%) that underwent WES analysis presented a multilocus genomic variation, in line with previous studies [42]. Notably, Patients 14 and 26 both present a variant in a gene associated with NSHL and a variant in a gene associated with Waardenburg syndrome, even if, from a clinical point of view, both subjects were classified as apparently NSHL cases. Waardenburg syndrome is characterised by HL together with pigmentation abnormalities of the skin, hair, and eyes, and other possible features, such as dystopia canthorum, skeletal abnormalities, or Hirschsprung disease [43]. Some of these clinical features might be subtle and, as already discussed, Waardenburg syndrome may also be a non-syndromic mimic. In this light, a thorough clinical evaluation of both patients and their fathers (from whom they inherited the variants) is planned in the following months. Conversely, both patients that were first diagnosed as SHL cases were found to be affected by NSHL together with a second genetic disorder. Indeed, Patient 46 presents *GJB2*-associated HL and *RPGR*-associated retinitis pigmentosa while Patient 50 is affected by *WFS1*-related HL and pseudoxanthoma elasticum due to an *ABCC6* mutation. In the first case, the patient was initially suspected to have a single condition that could explain all her clinical features, while the in the second patient the presence of two different genetic disorders was promptly hypothesised upon the clinical evaluation. These two cases underline how sometimes the clinical diagnosis of two genetic disorders is straightforward while in others only a molecular analysis can provide a definitive explanation of patients’ signs and symptoms.

## 5. Conclusions

In conclusion, this study underlines how the molecular landscape behind HHL is extremely complex, given the high genetic and clinical heterogeneity of this phenotype. Additionally, some peculiar scenarios are emerging, such as the occurrence of non-syndromic mimics and multilocus genomic variation, further challenging the achievement of a correct molecular diagnosis. In this context, a multidisciplinary approach that combines a comprehensive clinical evaluation together with a careful analysis of genomic data is the key to provide patients and their families with precise information about the prognosis, treatment, follow-up, and recurrence risks, thus representing the essential basis of a personalised clinical management.

## Figures and Tables

**Figure 1 biomedicines-11-00703-f001:**
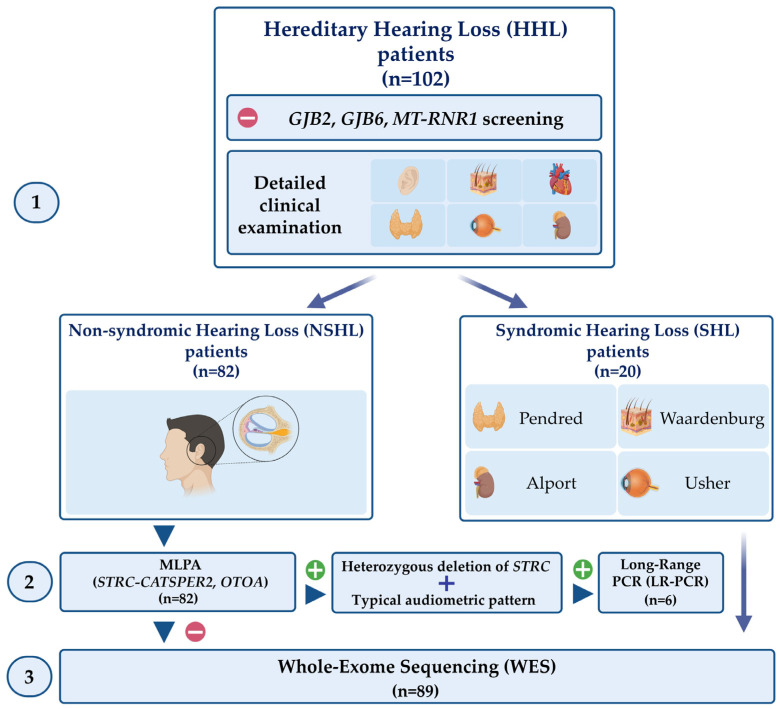
Schematic representation of the multistep and integrative approach performed in this study. One-hundred and two patients affected by HHL and negative in *GJB2*, *GJB6*, and *MT-RNR1* screening underwent a detailed clinical examination (1) aimed to differentiate NSHL patients from SHL ones. The patients affected by NHSL were screened for deletions in the *STRC-CATSPER2* and *OTOA* genes (2). For the NSHL patients carrying a heterozygous deletion of *STRC* with a typical audiometric pattern, *STRC* sequencing was performed. WES was carried out in all NSHL patients negative for these steps and in SHL patients. *Created with BioRender.com* (accessed on 25 January 2023).

**Figure 2 biomedicines-11-00703-f002:**
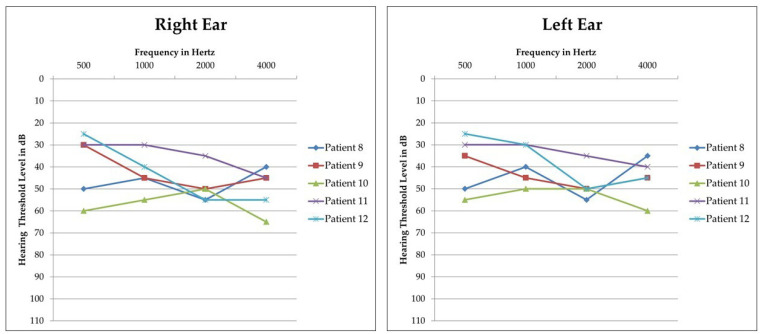
Audiograms of patients carrying a heterozygous *STRC* deletion and a hemizygous pathogenic or likely pathogenic in trans variant. Threshold of the right and left ear of each patient is reported separately. All patients display mild-to-moderate bilateral sensorineural HL with a symmetric pattern between the right and left ear. Patient 13 did not undergo PTA but only ABR, due to her age.

**Table 1 biomedicines-11-00703-t001:** *STRC* deletions identified through MLPA analysis. Gene: gene carrying the deletion. Type of deletion: entire-gene deletion or specific exon(s) deletion. Genotype: homozygous: the same variant is present on both alleles; compound heterozygous: two different variants are present on each allele. Inheritance: inheritance pattern of the deletion established after parental segregation.

Patient ID	Gene	Type of Deletion	Genotype	Inheritance
1	*STRC*	Entire gene	Homozygous	Paternal and Maternal
2	*STRC*	Entire gene	Homozygous	Paternal and Maternal
3	*STRC* *STRC*	Entire gene	Compound heterozygous	Paternal
		Exon 19		Maternal
4	*STRC*	Entire gene	Homozygous	Paternal and Maternal
5	*STRC*	Entire gene	Homozygous	Paternal and Maternal
6	*STRC*	Entire gene	Homozygous	Paternal and Maternal
7	*STRC*	Entire gene	Homozygous	Paternal and Maternal

**Table 2 biomedicines-11-00703-t002:** *STRC* deletions and SNVs identified in the six patients that underwent both MLPA and LR-PCR analyses. Gene: gene carrying the deletion or the identified variants with NCBI RefSeq accession number of the considered protein-coding transcripts (NM_). cDNA change and Protein change: variant description according to the Human Genome Variation Society (HGVS) nomenclature guidelines. Genotype: compound heterozygous: two different variants are present on each allele. Inheritance: inheritance pattern of every identified variant established after parental segregation. ACMG/AMP classification: variant pathogenicity according to ACMG/AMP guidelines. References: PubMed unique IDentifier (PMID) of publications reporting each variant. *: stop codon. NA: not available.

Patient ID	Gene	cDNA Change	Protein Change	Genotype	Inheritance	ACMG/AMP Classification	References
8	*STRC* (NM_153700.2)	Entire-gene deletion	NA	Compound heterozygous	Paternal	NA	PMID: 21686705
		c.1873C>T	p.Arg625Cys		Maternal	Uncertain significance	PMID: 22147502
9	*STRC* (NM_153700.2)	Entire-gene deletion	NA	Compound heterozygous	Paternal	NA	PMID: 21686705
		c.1873C>T	p.Arg625Cys		Maternal	Uncertain significance	PMID: 22147502
10	*STRC* (NM_153700.2)	Entire-gene deletion	NA	Compound heterozygous	Paternal	NA	PMID: 21686705
		c.4510delG	p.(Glu1504Argfs * 32)		Maternal	Pathogenic	PMID: 27068579
11	*STRC* (NM_153700.2)	Entire-gene deletion	NA	Compound heterozygous	Paternal	NA	PMID: 21686705
		c.4917_4918delACinsCT	p.(Leu1640Phe)		Maternal	Uncertain significance	PMID: 36086952
12	*STRC* (NM_153700.2)	c.2405_2407delTGT	p.(Leu802_Ser803 delinsPro)	Compound heterozygous	Paternal	Uncertain significance	NA
		Entire-gene deletion	NA		Maternal	NA	PMID: 21686705
13	*STRC* (NM_153700.2)	c.4402C>T	p.(Arg1468 *)	Compound heterozygous	Paternal	Pathogenic	PMID: 26011646NA
		Entire-gene deletion	NA		Maternal	NA	PMID: 21686705

**Table 3 biomedicines-11-00703-t003:** Variants identified through WES analysis in apparent NSHL patients. Gene: list of genes carrying the identified variants with NCBI RefSeq accession number of the considered protein-coding transcripts (NM_). Associated disease: genetic disorder nomenclature according to the Online Mendelian Inheritance in Man^®^ (OMIM^®^) database. Transmission pattern: AD: autosomal dominant; AR: autosomal recessive. cDNA change and Protein change: variant description according to the Human Genome Variation Society (HGVS) nomenclature guidelines. Genotype: homozygous: the same variant is present on both alleles; heterozygous: the variant affects only one allele; compound heterozygous: two different variants are present on each allele. Inheritance: inheritance pattern of every identified variant established after parental segregation. ACMG/AMP classification: variant pathogenicity according to ACMG/AMP guidelines. References: PubMed unique IDentifier (PMID) of publications reporting each variant. *: stop codon. NA: not available.

Patient ID	Gene	Associated Disease	Transmission Pattern	cDNA Change	Protein Change	Genotype	Inheritance	ACMG/AMP Classification	References
14	*ATP2B2* (NM_00168.3)	Deafness, autosomal dominant 82 (MIM:# 619804)	AD	c.1924 T>G	p.(Trp642Gly)	Heterozygous	Paternal	Uncertain significance	NA
	*EDN3* (NM_207034.1)	Waardenburg syndrome, type 4B (MIM: # 613265)	AD	c.167_190dupAGACTGTGGCTGGCCCTGGCGAGG	p.(Glu56_Glu63dup)	Heterozygous	Paternal	Uncertain significance	NA
15	*GATA3* (NM_001002295.1)	Hypoparathyroidism, sensorineural deafness, and renal dysplasia (MIM: # 146255)	AD	c.812C>T	p.(Ser271Leu)	Heterozygous	De novo	Likely pathogenic	NA
16	*TRIOBP* (NM_001039141.2)	Deafness, autosomal recessive 28 (MIM: # 609823)	AR	c.2827C>T	p.(Gln943 *)	Compound heterozygous	Paternal	Pathogenic	NA
				c.5014G>T	p.(Gly1672 *)		Maternal	Pathogenic	PMID: 29197352
17	*HARS2* (NM_012208.3)	Perrault syndrome 2 (MIM: # 614926)	AR	c.1273C>T	p.(Arg425Trp)	Compound heterozygous	Paternal	Uncertain significance	NA
				c.389A>G	p.(Leu1640Phe)		Maternal	Uncertain significance	NA
18	*OTOF* (NM_194248.2)	Auditory neuropathy, autosomal recessive, 1 (MIM: # 601071)	AR	c.4981G>A	p.(Glu1661Lys)	Compound heterozygous	Paternal	Uncertain significance	PMID: 36672845
				c.5533+13G>T	NA		De novo	Uncertain significance	NA
19	*TMPRSS3* (NM_024022.4)	Deafness, autosomal recessive 8/10 (MIM: # 601072)	AR	c.1224delA	p.(Glu409Argfs * 7)	Compound heterozygous	Paternal	Likely pathogenic	NA
				c.646C>T	p.(Arg216Cys)		Maternal	Pathogenic	PMID: 34440452
20	*LOXHD1* (NM_144612.6)	Deafness, autosomal recessive 77 (MIM: # 613079)	AR	c.4480C>T	p.(Arg1494 *)	Compound heterozygous	Paternal	Pathogenic	PMID: 25792669
				c.5085+970T>C	NA		Maternal	Uncertain significance	NA
21	*LRTOMT* (NM_001145309.3)	Deafness, autosomal recessive 63 (MIM: # 611451)	AR	c.358+4A>C	NA	Homozygous	Paternal and Maternal	Likely pathogenic	PMID: 18953341
22	*CEACAM16* (NM_001039213.2)	Deafness, autosomal dominant 4B (MIM: # 614614)	AD	c.505G>A	p.(Gly169Arg)	Heterozygous	NA	Uncertain significance	PMID: 25589040
23	*OTOF* (NM_194248.2)	Auditory neuropathy, autosomal recessive, 1 (MIM: # 601071)	AR	c.1699delG	p.(Ala567fs)	Compound heterozygous	Paternal	Pathogenic	NA
				c.1601delC	p.(Pro534fs)		Maternal	Pathogenic	PMID: 18381613
24	*OTOGL* (NM_173591.3)	Deafness, autosomal recessive 84B (MIM: # 614944)	AR	c.6754+4A>C	NA	Compound heterozygous	Paternal	Uncertain significance	NA
				c.448C>T	p.(Arg150Trp)		Maternal	Uncertain significance	NA
25	*MYH14* (NM_001145809.2)	Deafness, autosomal dominant 4A (MIM: # 600652)	AD	c.4088G>A	p.(Arg1363His)	Heterozygous	Maternal	Uncertain significance	NA
26	*COL11A1* (NM_001854.4)	Deafness, autosomal dominant 37 (MIM: # 618533)	AD	c.611C>A	p.(Thr204Lys)	Heterozygous	Paternal	Uncertain significance	NA
	*PAX3* (NM_000438.6)	Waardenburg syndrome, type 1 (MIM: # 193500)	AD	c.599G>A	p.(Arg200His)	Heterozygous	Paternal	Uncertain significance	NA
27	*CLDN14* (NM_001146079.2)	Deafness, autosomal recessive 29 (MIM: # 614035)	AR	c.664delG	p.(Ala222fs)	Compound heterozygous	Paternal	Pathogenic	PMID: 32747562
				c.467T>C	p.(Met156Thr)		Maternal	Uncertain significance	NA
28	*KCNQ4* (NM_004700.4)	Deafness, autosomal dominant 2A (MIM: # 600101)	AD	c.845C>A	p.(Thr282Lys)	Heterozygous	Paternal	Likely pathogenic	NA
29	*MYO15A* (NM_016239.4)	Deafness, autosomal recessive 3 (MIM: # 600316)	AR	c.1390delG	p.(Asp464Ilefs * 22)	Compound heterozygous	Paternal	Pathogenic	NA
				c.4777G>A	p.(Glu1593Lys)		Maternal	Uncertain significance	PMID: 32860223
30	*ADGRV1* (NM_032119.4)	Usher syndrome, type 2C (MIM: # 605472)	AR	c.9447+1G>A	NA	Compound heterozygous	Paternal	Pathogenic	NA
				c.13655dupT	p.(Asn4553Glufs * 18)		Maternal	Pathogenic	PMID: 33105617
31	*PLS1* (NM_001145319.2)	Deafness, autosomal dominant 76 (MIM: # 618787)	AD	c.542C>A	p.(Ala181Asp)	Heterozygous	Maternal	Uncertain significance	NA
32	*GJB2* (NM_004004.5)	Deafness, autosomal recessive 1A (MIM: # 220290)	AR	c.-22-2A>C	NA	Compound heterozygous	Paternal	Pathogenic	PMID: 34062854
				c.269T>C	p.(Leu90Pro)		Maternal	Likely pathogenic	PMID: 29293505
33	*ADGRV1* (NM_032119.4)	Usher syndrome, type 2C (MIM: # 605472)	AR	c.18907G>A	p.(Asp6303Asn)	Compound heterozygous	Paternal	Uncertain significance	NA
				c.17951A>C	p.(Gln5984Pro)		Maternal	Uncertain significance	NA
34	*MITF* (NM_001354604.2)	Waardenburg syndrome, type 2A (MIM: # 193510)	AD	c.1031+1G>A	NA	Heterozygous	De novo	Pathogenic	PMID: 32013026
35	*USH2A* (NM_206933.4)	Usher syndrome, type 2A (MIM: # 276901)	AR	c.9270C>A	p.(Cys3090 *)	Compound heterozygous	Paternal	Pathogenic	PMID: 35266249
				c.1876C>T	p.(Arg626 *)		Maternal	Pathogenic	PMID: 27460420
36	*USH2A* (NM_206933.4)	Usher syndrome, type 2A (MIM: # 276901)	AR	c.2099_2120delGGACAGTGGATGGAGATATTAC	p.(Gly700Alafs * 49)	Compound heterozygous	Paternal	Pathogenic	NA
				c.8167C>T	p.(Arg2723 *)		Maternal	Pathogenic	PMID: 19683999
37	*USH2A* (NM_206933.4)	Usher syndrome, type 2A (MIM: # 276901)	AR	c.6778T>C	p.(Ser2260Pro)	Compound heterozygous	Paternal	Uncertain significance	PMID: 20507924
				c.12067-2A>G	NA		Maternal	Pathogenic	PMID: 27460420
38	*KARS1* (NM_001130089.1)	Deafness, autosomal recessive 89 (MIM: # 613916)	AR	c.346A>G	p.(Lys116Glu)	Compound heterozygous	Paternal	Uncertain significance	NA
				c.1124A>G	p.(Tyr375Cys)		Maternal	Uncertain significance	PMID: 29615062
39	*STRC* (NM_153700.2)	Deafness, autosomal recessive 16 (MIM: # 603720)	AR	c.3511T>C	p.(Trp1171Arg)	Compound heterozygous	Paternal	Uncertain significance	NA
				c.4917_4918delACinsCT	p.(Leu1640Phe)		Maternal	Uncertain significance	PMID: 36086952

**Table 4 biomedicines-11-00703-t004:** Variants identified through WES analysis in SHL patients. Gene: list of genes carrying the identified variants with NCBI RefSeq accession number of the considered protein-coding transcripts (NM_). Associated disease: genetic disorder nomenclature according to the Online Mendelian Inheritance in Man^®^ (OMIM^®^) database. Transmission pattern: AD: autosomal dominant; AR: autosomal recessive; XL: X-linked. cDNA change and Protein change: variant description according to the Human Genome Variation Society (HGVS) nomenclature guidelines. Genotype: homozygous: the same variant is present on both alleles; heterozygous: the variant affects only one allele; compound heterozygous: two different variants are present on each allele. Inheritance: inheritance pattern of every identified variant established after parental segregation. ACMG/AMP classification: variant pathogenicity according to ACMG/AMP guidelines. References: PubMed unique IDentifier (PMID) of publications reporting each variant. *: stop codon. NA: not available.

Patient ID	Gene	Associated Disease	Transmission Pattern	cDNA Change	Protein Change	Genotype	Inheritance	ACMG/AMP Classification	References
40	*CHD7* (NM_017780.3)	CHARGE syndrome (MIM: # 214800)	AD	c.550C>T	p.(Gln184 *)	Heterozygous	De novo	Pathogenic	PMID: 16615981
41	*USH2A* (NM_206933.2)	Usher syndrome, type 2A (MIM: # 276901)	AR	c.11713C>T	p.(Arg3905Cys)	Compound heterozygous	Paternal	Pathogenic	PMID: 15015129
				c.11864G>A	p.(Trp3955 *)		Maternal	Pathogenic	PMID: 31877679
42	*SF3B2* (NM_006842.2)	Craniofacial microsomia (MIM: # 164210)	AD	c.1660C>T	p.(Arg554 *)	Heterozygous	Maternal	Likely pathogenic	NA
43	*EYA1* (NM_000503.4)	Branchiootorenal syndrome 1 (MIM: # 113650)	AD	c.880C>T	p.(Arg294 *)	Heterozygous	De novo	Pathogenic	PMID: 21280147
44	*CDH23* (NM_022124.6)	Usher syndrome, type 1D (MIM: # 601067)	AR	c.9433C>T	p.(Gln3145 *)	Compound heterozygous	Paternal	Pathogenic	NA
				c.5712G>A	p.(Thr1904Thr)		Maternal	Likely pathogenic	PMID: 21738395
45	*ADGRV1* (NM_032119.4)	Usher syndrome, type 2C (MIM: # 605472)	AR	c.12226_12227del ATinsGTAGAT GAGAGTAGATG	p.(Ile4076_Leu6306delinsValAspGluSerArg)	Compound heterozygous	Paternal	Pathogenic	NA
				c.11410C>T	p.(Arg3804 *)		Maternal	Pathogenic	PMID: 28944237
46	*GJB2* (NM_04004.5)	Deafness, autosomal recessive 1A (MIM: # 220290)	AR	c.35delG	p.(Gly12Valfs * 2)	Homozygous	Paternal and Maternal	Pathogenic	PMID: 9620796
	*RPGR* (NM_000328.3)	Retinitis pigmentosa 3 (MIM: # 300029)	XL	c.591_592delTG	p.(Cys197Trpfs * 13)	Heterozygous	De novo	Pathogenic	NA
47	*PPP1R12A* (NM_002480.3)	Genitourinary and/or/brain malformation syndrome (MIM: # 618820)	AD	c.792+3A>C	NA	Heterozygous	De novo	Uncertain significance	NA
48	*PCDH15* (NM_001384140.1)	Usher syndrome, type 1F (MIM: # 602083)	AR	c.1737C>G	p.(Tyr579 *)	Compound heterozygous	NA	Pathogenic	PMID: 22815625
				Gene deletion			NA	Pathogenic	NA
49	*CDH23* (NM_022124.6)	Usher syndrome, type 1D (MIM: # 601067)	AR	c.5985C>A	p.(Tyr1995 *)	Homozygous	Paternal and Maternal	Pathogenic	PMID: 20513143
50	*WFS1* (NM_006005.3)	Deafness, autosomal dominant 6/14/38 (MIM: # 600965)	AD	c.2339G>C	p.(Gly780Ala)	Heterozygous	Maternal	Likely pathogenic	PMID: 23981289
	*ABCC6* (NM_001171.6)	Pseudoxanthoma elasticum (MIM: # 264800)	AR	c.3491G>A	p.(Arg1164Gln)	Homozygous	Paternal and Maternal	Likely pathogenic	PMID: 16086317

## Data Availability

Data presented in this study are available upon request to the corresponding author. Data are not publicly available due to privacy restrictions.

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
