# Peer review of "The Enigmatic Genetic Landscape of Hereditary Hearing Loss: A Multistep Diagnostic Strategy in the Italian Population"

_biomedicines, 2023, doi:10.3390/biomedicines11030703_

Round 1

Reviewer 1 Report

The molecular-genetic investigation in hearing impaired patients is very actual. The whole impression of the manuscript is good but it needs corrections to highlight the main subject. 

2.7, 3.4 sections: it seems that the carrier rate of pathogenic variants in most frequent genes associated with hearing loss could be the subject of another article and include not only USH2A and ADGRV1 genes but GJB2 and STRC as well. They are more frequent but no information of their carrier rates in the Italian population was provided in the presented manuscript.

Results: the first paragraph and Figure 1 are more correct to be placed in materials and methods chapter after clinical evaluation. 

Tables 1-3 – please leave only the first sentence as a table name and place explanation below the table or in the text.

Lines 290-302 – the data could be presented in absolute numbers only due to not numerous sample size (26 subjects).

Line 323 – incorrect to mention the Waardenburg syndrome type 3 here, because it is caused by PAX3 variant in homozygous state, but is this case only one variant was found.

Line 462 - Patients with USH2A pathogenic variants are mostly good hearing aid users, the CI decision should be based only on hearing status

Author Response

We have taken into considerations all the Reviewer’s comments and hope to have addressed all question clearly.

Reviewer #1 (Comments to the Author):
The molecular-genetic investigation in hearing impaired patients is very actual. The whole impression of the manuscript is good but it needs corrections to highlight the main subject.

1. 2.7, 3.4 sections: it seems that the carrier rate of pathogenic variants in most frequent genes associated with hearing loss could be the subject of another article and include not only USH2A and ADGRV1 genes but GJB2 and STRC as well. They are more frequent but no information of their carrier rates in the Italian population was provided in the presented manuscript.

We thank the Reviewer for the comment. In this study cohort, we noticed a peculiar enrichment in patients affected by Usher syndrome type 2. In particular, the prevalence of Usher syndrome type 2 patients in our cohort was slightly higher than previously reported in the literature (PMID: 20301442). Notably, no published study has so far estimated the prevalence of USH2A and ADGRV1 pathogenic variants carriers in the Italian population. In this light, we were specifically interested in evaluating of pathogenic variants carriers rate in Usher syndrome type 2-associated genes in the Italian population, and not in the carriers’ rate of pathogenic variants in the most frequent hearing loss (HL) genes.

The frequency of pathogenic variants carriers within HL genes is certainly an intriguing aspect to be deepened. However, since this study aimed to provide a detailed characterisation of the genetic causes of hereditary HL other than GJB2 mutations, the evaluation of GJB2 pathogenic variants carriers is not in line with our manuscript’s main goal. Regarding STRC, in this study, the prevalence of HL patients carrying homozygous deletions within this gene is not statistically significantly higher than reported in the literature (PMID: 30867468) (Exact Binomial Test: p-value>0.05). For this reason, we did not evaluate the prevalence of STRC pathogenic variants carriers in the Italian population since it would not provide any novelty compared with the already available literature data. 2.

2. Results: the first paragraph and Figure 1 are more correct to be placed in materials and methods chapter after clinical evaluation.

We acknowledge the Reviewer for comment. To make clear the multistep approach performed in this study, we modified the 2.2 Clinical evaluation into 2.2 Study design paragraph in the Materials and Methods section (Line 95). In particular, we reported a brief summary of the steps performed, as follows: Lines 103-108: “Enrolled subjects were analysed through a multistep approach that comprises the following steps: 1) a detailed clinical evaluation to distinguish NSHL patients from SHL ones; 2) MLPA analysis of STRC-CATSPER2 and OTOA in NSHL patients; 3) LR-PCR and STRC sequencing in patients carrying a heterozygous deletion of STRC and an audiometric pattern suggestive of Deafness, autosomal recessive 16; 4) WES analysis in patients negative to steps 2 and 3 and in SHL subjects”.

Regarding Figure 1, we believe that it is more functional for readers in the Results section. Indeed, Figure 1 provides not only the study flowchart, but also the total number of the enrolled participants and the number of patients affected by NSHL and SHL. To make the figure more comprehensive, we implemented it with the number of patients considered in each analysis step.

3. Tables 1-3 – please leave only the first sentence as a table name and place explanation below the table or in the text.

We thank the Reviewer for the kind suggestion. However, the formatting of the table name and relative description is due to the fact that we followed Biomedicines guidelines, and we wrote the manuscript’s text in Biomedicines predisposed template.

4. Lines 290-302 – the data could be presented in absolute numbers only due to not numerous sample size (26 subjects). 

We acknowledge the Reviewer for the comment. We modified this section presenting only the absolute numbers (please, see Lines 325, 332, 335).

5. Line 323 – incorrect to mention the Waardenburg syndrome type 3 here, because it is caused by PAX3 variant in homozygous state, but is this case only one variant was found.

We thank the Reviewer for the comment. Waardenburg Syndrome type 3 (WS-III) is caused by heterozygous or homozygous mutations in the PAX3 gene. In literature, there are some reports that provide evidence of an autosomal dominant inheritance of WS-III, such as:

• The study performed by Sheffer et al. (PMID: 1536170) is the first literature report regarding the autosomal-dominant inheritance of WS-III.

• Tekin et al. (PMID: 11683776) reported another example of a family (mother and son) with typical clinical findings of WS-III segregating with a heterozygous deletion in the PAX3 gene.

Considering the possibility of autosomal dominant inheritance of WS-III, in the Results section we stated that heterozygous variants in the PAX3 gene may cause both Waardenburg syndrome type 1 and type 3. However, since the patient we describe does not show any upper limb abnormalities, for clarity reasons, we amended the description and only left that PAX3 may be associated with Waardenburg syndrome type 1 (please, see Lines 361-362).

6. Line 462 - Patients with USH2A pathogenic variants are mostly good hearing aid users, the CI decision should be based only on hearing status.

We acknowledge the Reviewer for the comment. Indeed, in patients affected by Usher syndrome type 2, auditory rehabilitation usually starts in early childhood with the use of hearing aids. However, in cases with severe progressive hearing loss that affects speech discrimination and causes communication difficulties, cochlear implants have proven to be effective (PMID: 34209904). In this light, we modified the Discussion section as follows (please, see Lines 503-504): “whenever patients develop severe progressive HL with poor speech discrimination and communication difficulties even with hearing aids, being these children excellent cochlear implant candidates, surgery should be performed before the onset of ocular problems to maximise communication skills”.

We hope that the revised version of our manuscript will be now suitable for acceptance in Biomedicines.

Reviewer 2 Report

Review

to manuscript “The enigmatic genetic landscape of Hereditary Hearing Loss: a multistep diagnostic strategy in the Italian population” by Beatrice Spedicati, Aurora Santin, Giuseppe Giovanni Nardone *, Elisa Rubinato, Stefania Lenarduzzi, Claudio Graziano, Livia Garavelli, Sara Miccoli, Stefania Bigoni, Anna Morgan, Giorgia Girotto

This study focused in the genetic landscape of the hereditary hearing loss: a multistep diagnostic strategy in the Italian population. In overall, this is a good job. However, I have some comments and recommendations.

Major comments

1.        The Result section.  The section about STRC&OTOA variants is not very clear; I can’t understand what was found in patients?  What deletion was found in the homozygous state? What variants in the compound heterozygous and what in the single heterozygous? I think the table 1 is not organized informatively. It would be better if this table included all patients including homozygotes and heterozygous patients (not only heterozygotes).  Also, there appear to be typos. For example, in the text authors write … “Parental segregation of the deletion showed that it had been inherited from the father in four cases and from the mother in the remaining two”. However in the table 1, on the contrary in two patients paternal inheritance and other cases mother inheritance. I think the authors should make this section as clear as possible (exactly which options in which condition and how many patients it were found); Authors should carefully check everything in this part of the article.

2.        The Discussion section. Since the authors present audiological data in patients with STRC variants (although it is not clear with which ones): I think this section needs to be improved. In my knowledge the genotype-phenotypic analysis of STRC gene has not been studied in large cohort of patients, and in this case the authors should more discuss about finding clinical features in comparison with previously obtained genotype-phenotypic works in discussion section. In present variant of the manuscript the not all recent works about genotype-phenotype correlation are cited by the authors, for example:

-          T G Markova, N N Alekseeva, O L Mironovich, N M Galeeva, M R Lalayants, E A Bliznetz, S S Chibisova, A V Polyakov , G A Tavartkiladze Clinical features of hearing loss caused by STRC gene deletions/mutations in Russian population Int J Pediatr Otorhinolaryngol . 2020 Nov;138:110247. doi:10.1016/j.ijporl.2020.110247. Epub 2020 Jul 19.

-          Pavlina Plevova, Martina Paprskarova, Petra Tvrda, Petra Turska, Rastislav Slavkovsky, Eva Mrazkova STRC Deletion is a Frequent Cause of Slight to Moderate Congenital Hearing Impairment in the Czech Republic Otol Neurotol . 2017 Dec;38(10):e393-e400. doi: 10.1097/MAO.0000000000001571.

-          Andrea Simi, Julia Perry, Emma Schindler, Andrea Oza, Minjie Luo, Tiffiney Hartman, Ian D Krantz, John A Germiller, Kosuke Kawai, Margaret Kenna. Audiologic Phenotype and Progression in Pediatric STRC-Related Autosomal Recessive Hearing Loss Laryngoscope. 2021 Dec;131(12):E2897-E2903. doi: 10.1002/lary.29680. Epub 2021 Jun 10.

Minor comments

1. The title. I think the title of manuscript need to be corrected. For example “The genetic landscape of Hereditary Hearing Loss: a multistep diagnostic strategy in the Italian population”;

2. The introduction section. What mean the HL in introduction section? I think full description of HL is missing.

3. The Result section.  The authors should write more clearly that audiological analysis of the STRC-heterozygote’s was available for 5 of 6 patients.

Recommendation

Accept after major comments

Author Response

We have taken into considerations all the Reviewer’s comments and hope to have addressed all question clearly.

Reviewer #2 (Comments to the Author):
This study focused in the genetic landscape of the hereditary hearing loss: a multistep diagnostic strategy in the Italian population. In overall, this is a good job. However, I have some comments and recommendations.

Major comments

1. The Result section. The section about STRC&OTOA variants is not very clear; I can’t understand what was found in patients? What deletion was found in the homozygous state? What variants in the compound heterozygous and what in the single heterozygous? I think the table 1 is not organized informatively. It would be better if this table included all patients including homozygotes and heterozygous patients (not only heterozygotes). Also, there appear to be typos. For example, in the text authors write … “Parental segregation of the deletion showed that it had been inherited from the father in four cases and from the mother in the remaining two”. However in the table 1, on the contrary in two patients paternal inheritance and other cases mother inheritance. I think the authors should make this section as clear as possible (exactly which options in which condition and how many patients it were found); Authors should carefully check everything in this part of the article.

We acknowledge the Reviewer for the comment. We better specified in the Materials and Methods section the exons covered by MLPA probes in order to precisely indicate the type of deletions we identified. Please, see Lines 135-142 and Lines 148-152: “The kit contains different probes spanning through the aforementioned genes. In particular, seven probes for the STRC gene (covering exons 19, 20, 23, 24, 25, and 28), five probes for the CATSPER2 gene (covering exons 1, 2, 4, and 7), and ten probes for the OTOA gene (covering exons 2, 5, 7, 8, 11, 12, 16, 17, 18, and 20) are present. Multiple flanking probes are also included in the 15q15.3 and 16q12.2 regions, indicating the extent of possible Copy Number Variations (CNVs). Furthermore, four STRC pseudogene (p-STRC) probes are present, to identify possible gene conversions. […] The following cut-offs for dosage quotient (DQ) of the probes were applied to interpret MLPA results: 0.80 < DQ < 1.20 (no deletion/duplication), DQ = 0 (homozygous deletion), 0.40 < DQ < 0.65 (heterozygous deletion), 1.30 > DQ > 1.65 (heterozygous duplication) and 1.75 < DQ < 2.15 (homozygous duplication/heterozygous triplication)”.

We also modified the Results section according to the Reviewer’s suggestions to make it as clear as possible. In particular, we added a table with its own caption (new Table 1) that reports the details of the six patients affected by non-syndromic hearing loss that carry a homozygous deletion of the STRC gene (please, see Page 6). We also modified the text accordingly at Lines 260-264: “Six patients (7.3%) carried a homozygous deletion of the entire STRC gene and one patient (Patient 3) presented an entire STRC gene deletion in compound heterozygosity with a deletion involving only exon 29 of the same gene; none of the analysed subjects presented a biallelic deletion involving OTOA (Table 1)”.

Additionally, we modified Table 2 in order to make it more informative on the compound heterozygous patients carrying both an SNV and a deletion of the STRC gene. For each patient, we added the inheritance pattern of the heterozygous deletion so that it would be straightforward to understand from which parent the deletion and the SNV were inherited, thus underlining their in trans presence and confirming their role in determining the clinical phenotype of the patients. Accordingly, we modified the caption of Table 2.

In this way, we hope that it appears more explicit that four patients inherited the deletion from the father and the SNV from the mother, whereas two of them inherited the deletion from the mother and the SNV from the father. Please see Lines 272-274 and Lines 284-287: “Parental segregation of the deletion showed that it had been inherited from the father in four cases and from the mother in the remaining two. […] In all of them, a hemizygous variant was identified and parental segregation showed that it had been maternally inherited in four if them and paternally in the last two cases, thus confirming their presence, in trans, with the previously identified gene deletion”.

2. The Discussion section. Since the authors present audiological data in patients with STRC variants (although it is not clear with which ones): I think this section needs to be improved. In my knowledge the genotype-phenotypic analysis of STRC gene has not been studied in large cohort of patients, and in this case the authors should more discuss about finding clinical features in comparison with previously obtained genotype-phenotypic works in discussion section. In present variant of the manuscript the not all recent works about genotype-phenotype correlation are cited by the authors, for example:

  • T G Markova, N N Alekseeva, O L Mironovich, N M Galeeva, M R Lalayants, E A Bliznetz, S S Chibisova, A V Polyakov , G A Tavartkiladze Clinical features of hearing loss caused by STRC gene deletions/mutations in Russian population Int J Pediatr Otorhinolaryngol . 2020 Nov;138:110247. doi:10.1016/j.ijporl.2020.110247. Epub 2020 Jul 19.
  • Pavlina Plevova, Martina Paprskarova, Petra Tvrda, Petra Turska, Rastislav Slavkovsky, Eva Mrazkova STRC Deletion is a Frequent Cause of Slight to Moderate Congenital Hearing Impairment in the Czech Republic Otol Neurotol . 2017 Dec;38(10):e393-e400. doi: 10.1097/MAO.0000000000001571.
  • Andrea Simi, Julia Perry, Emma Schindler, Andrea Oza, Minjie Luo, Tiffiney Hartman, Ian D Krantz, John A Germiller, Kosuke Kawai, Margaret Kenna. Audiologic Phenotype and Progression in Pediatric STRC-Related Autosomal Recessive Hearing Loss Laryngoscope. 2021 Dec;131(12):E2897-E2903. doi: 10.1002/lary.29680. Epub 2021 Jun 10.
  •  

We thank the Reviewer for the suggestion. We implemented the Discussion section highlighting the fact that the genotype-phenotype correlation between STRC loss-of function and mild-to-moderate hearing loss has been reported in studies involving a limited number of individuals. Please, see Lines 459-463: “Despite the fact that a detailed genotype-phenotype correlation between STRC loss-of-function and mild-to-moderate HL has been reported in studies involving a limited number of participants, this correspondence seems extremely consistent in all of them”.

We added two of the references suggested by the Reviewer (PMID: 32705992, PMID: 34111299) and two additional references (PMID: 31645979, PMID: 32203226). Anyhow, despite the limited number of reported patients, the genotype-phenotype correlation appears consistent, since, in all the above-mentioned published papers, subjects affected by Deafness, autosomal recessive 16 present a mild-to-moderate hearing loss. In line with the literature, all patients affected by STRC-associated hearing loss in our cohort show a mild-to-moderate deafness.

Minor comments

1. The title. I think the title of manuscript need to be corrected. For example “The genetic landscape of Hereditary Hearing Loss: a multistep diagnostic strategy in the Italian population”.

We acknowledge the Reviewer for the comment. We conceived the title with the adjective “enigmatic” because we believe it strongly underlines the genetic complexity and heterogeneity underlying hearing loss in the Italian population. Indeed, along with some genes being major players in our population as well as in other ethnicities (i.e. GJB2 and STRC), we also report on some peculiar findings of our cohort. For instance, we identified a patient affected by Perrault syndrome and carrying two compound heterozygous variants in the HARS2 gene: approximately 100 patients with this disease have been reported to date (PMID: 25254289), therefore our report could be valuable to further implement the knowledge on this very rare disorder. Additionally, a particularly interesting finding of our study, as reported in the Results and Discussion sections, is represented by the high frequency of patients affected by Usher syndrome type 2. This has led us to evaluate for the first time the prevalence of USH2A and ADGRV1 pathogenic variants carriers in the Italian population, highlighting how it resulted higher than the one reported in previous studies (PMID: 20301515). Overall, we believe that the genetic landscape of hearing loss in the Italian population presents its own distinctive characteristics, which led us to draft the title of our manuscript as presented.

2. The introduction section. What mean the HL in introduction section? I think full description of HL is missing.

We thank the Reviewer for the comment. Indeed, we missed to fully explain the “HL” acronym in the Introduction section. We added the terms “hearing loss” next to the acronym “HL” at Line 50.

3. The Result section. The authors should write more clearly that audiological analysis of the STRC-heterozygote’s was available for 5 of 6 patients.

We thank the Reviewer for the suggestion. We better specified this aspect at Lines 274-279: “Five of them (Patients 8, 9, 10, 11, and 12) had previously undergone PTA and their bone conduction audiograms are reported in Figure 2: they all and presented an audiometric pattern compatible with Deafness, autosomal recessive 16 (MIM: # 603720), which is characterised by mild-to-moderate bilateral hearing impairment [4]. ABR test had been performed on one patient (Patient 13) due to her age and revealed a threshold compatible with a moderate HL”.

We hope that the revised version of our manuscript will be now suitable for acceptance in Biomedicines.

Round 2

Reviewer 1 Report

Authors have revised the manuscript according the reviewer's comments and suggestions.

Author Response

We have taken into considerations the Reviewer’s comment:
Reviewer #1 (Comments to the Author):

Authors have revised the manuscript according the reviewer's comments and suggestions.

We acknowledge the Reviewer for all the provided comments and suggestions that significantly contributed to improve our manuscript’s quality and clarity.

We hope that the revised version of our manuscript will be now suitable for acceptance in Biomedicines.

Reviewer 2 Report

Review Round-2

to manuscript “The enigmatic genetic landscape of Hereditary Hearing Loss: a multistep diagnostic strategy in the Italian population” by Beatrice Spedicati, Aurora Santin, Giuseppe Giovanni Nardone *, Elisa Rubinato, Stefania Lenarduzzi, Claudio Graziano, Livia Garavelli, Sara Miccoli, Stefania Bigoni, Anna Morgan, Giorgia Girotto

The authors have improved the information content of the data presented in the section on STRC and OTOA variants. Also, authors added more additional information for make presented data for more clearly. The authors have corrected almost all my major and minor comments.  I have only one minor comment:

Minor comment

1. Although, the author's note of the type of inheritance available in the text of the manuscript, for more clearly, it would be better if they added information about type of inheritance in the Table 3 and Table 4.

Recommendation

Accept after minor comment

Author Response

We have taken into considerations all the Reviewer’s comments and hope to have addressed all question clearly.

Reviewer #2 (Comments to the Author): The authors have improved the information content of the data presented in the section on STRC and OTOA variants. Also, authors added more additional information for make presented data for more clearly. The authors have corrected almost all my major and minor comments. I have only one minor comment:

Minor comment
1. Although, the author's note of the type of inheritance available in the text of the manuscript, for more clearly, it would be better if they added information about type of inheritance in the Table 3 and Table 4.

We thank the Reviewer for the suggestion. We implemented Table 3 and Table 4 adding two columns. In one column we included for each gene the associated genetic disorder, providing the nomenclature according to the Online Mendelian Inheritance in Man® (OMIM®) database. In the other column we added, for each condition, the associated transmission pattern. This way, we hope to have better explained all the non-syndromic and syndromic conditions detected in our patients.

We hope that the revised version of our manuscript will be now suitable for acceptance in Biomedicines.
